# Near-Field IPO for Analysis of EM Scattering from Multiple Hybrid Dielectric and Conductor Target and High Resolution Range Profiles

**Qingkuan Wang** [1,†], **Yijin Wang** [1,*,†], **Chuangming Tong** [1], **Zhaolong Wang** [1], **Ximin Li** [2] **and Tong Wang** [1]

[1]   Air Defense and Antimissile School, Air Force Engineering University, Xi'an 710051, China
[2]   National Laboratory of Radar Signal Processing, Xidian University, Xi'an 710071, China
*   Correspondence: wang_yijin1021@163.com
†   These authors contributed equally to this work.

**Abstract:** Aiming at improving the accuracy and efficiency of scattering information from multiple targets in near-field regions, this paper proposes a near-field iterative physical optics (IPO) method based on a modified near-field Green's function for the composite electromagnetic scattering analysis of multiple hybrid dielectric and conductor targets. According to the electric field and magnetic field integral equation, the electric and magnetic current were updated utilizing the Jacobi iteration method. Then, by introducing an expansion center lying in the neighborhood of the source point, Green's function was modified for near-field scattering between multiple hybrid dielectric and conductor targets. To accelerate the implementation of the procedure, the multilevel fast multipole method, the fast far-field approximation, and parallel multicore programming were introduced. Numerical results indicate that there is good agreement between the results calculated by the near-field IPO method and MLFMM solver in commercial software FEKO while significantly reducing the computational burden. To fully exploit the scattering information, the high resolution range profiles (HRRP) of different targets under different conditions were analyzed, which can be further applied for automatic target detection and recognition.

**Keywords:** composite electromagnetic scattering; hybrid dielectric and conductor target; multiple targets; near-field IPO; high resolution range profiles

## 1. Introduction

The electromagnetic (EM) scattering analysis of rough surfaces and multiple hybrid dielectric and conductor targets is significant in air defense and antimissile applications [1]. Meanwhile, numerous innovative types of research have been conducted under different conditions. However, the studies on this topic mainly focused on the far-field scattering analysis of a single perfect electric conductor (PEC) target above a rough dielectric surface [2]. The coupling effect of multiple hybrid dielectric and conductor targets in the near field was not considered.

An efficient approximation strategy of near-field physical optics (PO) scattering from PEC objects illuminated by arbitrary far-field sources is presented in [3]. Similarly, based on PO approximation, the iterative physical optics (IPO) method has been utilized to iteratively calculate the interaction among complicated-shaped targets [4,5]. Moreover, an algorithm based on PO, physical theory of diffraction, and geometrical optics methods is proposed to calculate near-field RCS and echo signals in missile–target encounter simulations [6–9]. To obtain accurate near-field results, a PO near-field integral expression in the near zone is proposed by introducing locally expanded Green's function approximations [10]. A novel approach utilizing the PO-based near-field method is introduced to design wideband doubly-curved shaped reflector antennas [11]. A near-field series expansion of Green's function was utilized for the analysis of interconnects in layered media [12]. Analytic

formulas for Weakly Singular Integrals were derived and utilized to correct the near-field iteration in an IPO scheme [13]. In [14], a near-field shooting and bouncing ray based on PO is presented to analyze the scattering of the electrically large target in the near-field zone.

As for the analysis of composite scattering, combining the method of moment and PO, a new method is proposed to analyze composite scattering from multiple objects in [15]. To calculate the near-field scattering of electrically large PEC and dielectric coated objects, an improved IPO algorithm that considers the impedance boundary condition is presented in [16]. The acceleration method for backscattering computation in the near field and far field was proposed in [14] to accelerate the calculation procedure.

However, all the previous work has considered a single PEC target or one coated with dielectric materials. With the progress of material technology, dielectric materials are more and more widely used, and the electromagnetic scattering analysis of hybrid dielectric and conductor targets is more complex. Based on the near-field IPO method for multiple PEC targets, which has been studied by our research team [17], we modified near-field Green's function and present a near-field IPO method that is suitable for the composite EM scattering analysis of rough surface and multiple hybrid dielectric and conductor targets above it in the near-field region. We present the iterative physical optics expression and near-field correction; moreover, the acceleration method is introduced in Section 2. In Section 3, various numerical results are presented, and Section 4 presents the conclusions.

## 2. Methods

### 2.1. Iterative Physical Optics

According to the electric field and magnetic field integral equation, the principal value integral (P.V.) is described as

$$
\begin{aligned}
\boldsymbol{E}(\boldsymbol{r}) = \ & 2\boldsymbol{E}^{inc}(\boldsymbol{r}) + 2\text{P.V.}\int_S \{-j\omega\mu_0[\boldsymbol{J}(\boldsymbol{r}')G(\boldsymbol{r},\boldsymbol{r}') + (1/k_0^2)\nabla'\cdot\boldsymbol{J}(\boldsymbol{r}')\nabla G(\boldsymbol{r},\boldsymbol{r}')] + \\
& \boldsymbol{M}(\boldsymbol{r}')\times\nabla G(\boldsymbol{r},\boldsymbol{r}')\}\mathrm{d}S', \boldsymbol{r}\in S
\end{aligned}
\tag{1}
$$

$$
\begin{aligned}
\boldsymbol{H}(\boldsymbol{r}) = \ & 2\boldsymbol{H}^{inc}(\boldsymbol{r}) + 2\text{P.V.}\int_S \{-j\omega\varepsilon_0[\boldsymbol{M}(\boldsymbol{r}')g(\boldsymbol{r},\boldsymbol{r}') + (1/k_0^2)\nabla'\cdot\boldsymbol{M}(\boldsymbol{r}')\nabla G(\boldsymbol{r},\boldsymbol{r}')] - \\
& \boldsymbol{J}(\boldsymbol{r}')\times\nabla G(\boldsymbol{r},\boldsymbol{r}')\}\mathrm{d}S', \boldsymbol{r}\in S
\end{aligned}
\tag{2}
$$

where $S$ is the illuminated areas on the surface; $k_0$ represents the wavenumber in free space; $\boldsymbol{J}$ and $\boldsymbol{M}$ are the induced electric and magnetic current on the scatterer surface, respectively; $\boldsymbol{r}$ and $\boldsymbol{r}'$ denote observation and source points; $G$ is the free space Green's function, which can be described as [16]

$$
G\left(\boldsymbol{r},\boldsymbol{r}'\right) = \frac{\exp\left(jk|\boldsymbol{r}-\boldsymbol{r}'|\right)}{4\pi|\boldsymbol{r}-\boldsymbol{r}'|}.
\tag{3}
$$

Based on (1) and (2), $\boldsymbol{J}_i$ and $\boldsymbol{M}_i$ are presented as [18]

$$
\begin{aligned}
\boldsymbol{J}_i = \ & \boldsymbol{J}_i^{inc} + 2\sum_{j=1,i\neq j}^{N}\hat{\boldsymbol{n}}_i\times\Big\{\text{P.V.}\int_S\big\{-j\omega\varepsilon_0[\boldsymbol{M}_j(\boldsymbol{r}')G(\boldsymbol{r},\boldsymbol{r}') + \\
& (1/k_0^2)\nabla'\cdot\boldsymbol{M}_j(\boldsymbol{r}')\nabla G(\boldsymbol{r},\boldsymbol{r}')] - \boldsymbol{J}_j(\boldsymbol{r}')\times\nabla G(\boldsymbol{r},\boldsymbol{r}')\big\}\mathrm{d}S', \boldsymbol{r}\in S\Big\}
\end{aligned}
\tag{4}
$$

$$
\begin{aligned}
\boldsymbol{M}_i = \ & \boldsymbol{M}_i^{inc} + 2\sum_{j=1,i\neq j}^{N}(-\hat{\boldsymbol{n}}_i)\times\Big\{\text{P.V.}\int_S\big\{-j\omega\varepsilon_0[\boldsymbol{J}_j(\boldsymbol{r}')G(\boldsymbol{r},\boldsymbol{r}') + \\
& (1/k_0^2)\nabla'\cdot\boldsymbol{J}_j(\boldsymbol{r}')\nabla G(\boldsymbol{r},\boldsymbol{r}')] + \boldsymbol{M}_j(\boldsymbol{r}')\times\nabla G(\boldsymbol{r},\boldsymbol{r}')\big\}\mathrm{d}S', \boldsymbol{r}\in S\Big\}
\end{aligned}
\tag{5}
$$

where, $J_i$ and $M_i$ denote the induced electric and magnetic current on the facet $i$, while $J_i^{inc}$ and $M_i^{inc}$ stand for induced electric and magnetic current induced by the incident electric field and magnetic field, respectively, which are given by

$$\begin{aligned} J_i\left(r\right) &= \hat{n}_i \times H_i(r) \, , J_i^{inc}\left(r\right) = 2\hat{n}_i \times H_i^{inc}(r) \\ M_i(r) &= -\hat{n}_i \times E_i(r) \, , M_i^{inc}(r) = -2\hat{n}_i \times E_i^{inc}(r). \end{aligned} \tag{6}$$

Targets tend to be in the far field of the antennas. Therefore, the incident EM wave can be seen as the plane wave:

$$E^{inc}\left(r'\right) = \hat{e}^{inc} E^{inc} \exp\left(-jk\hat{k}^{inc}\cdot r'\right), H^{inc}\left(r'\right) = \frac{1}{Z}\hat{k}^{inc} \times E^{inc}\left(r'\right) \tag{7}$$

where $\hat{e}_i$ and $E_i$ are the polarization and amplitudes of the incident electric field, respectively.

By substituting (7) into (6), the iterative process starts with the initial conditions as follows:

$$J_i^0(r') = \frac{2E^{inc}}{Z}\hat{n} \times \left(\hat{k}^{inc} \times \hat{e}^{inc}\right) \exp\left(-jk\hat{k}^{inc}\cdot r'\right) \tag{8}$$

$$M_i^0(r') = -2\hat{n} \times \hat{e}^{inc} E^{inc} \exp\left(-jk\hat{k}^{inc}\cdot r'\right). \tag{9}$$

Furthermore, to update the current on the facet $i$, where $i = 1, 2, \ldots, N$, the Jacobi iteration algorithm is utilized; then, the surface currents can be represented as [17]

$$\begin{aligned} J_i^{(k+1)} = \ & J_i^{inc} + 2\sum_{j=1, i\neq j}^{N} \hat{n}_i \times \Big\{ \text{P.V.}\int_S \Big\{ -j\omega\varepsilon_0 [M_j^{(k)}(r')G(r,r') + (1/k_0^2)\nabla' \cdot M_j^{(k)}(r')\nabla G(r,r')] \\ & -J_j^{(k)}(r') \times \nabla G(r,r')\Big\}dS'\Big\}, r \in S \end{aligned} \tag{10}$$

$$\begin{aligned} M_i^{(k+1)} = \ & M_i^{inc} + 2\sum_{j=1, i\neq j}^{N} (-\hat{n}_i) \times \Big\{ \text{P.V.}\int_S \Big\{ -j\omega\varepsilon_0 [J_j^{(k)}(r') G(r,r') + (1/k_0^2)\nabla' \cdot J_j^{(k)}(r')\nabla G(r,r')] \\ & +M_j^{(k)}(r') \times \nabla G(r,r')\Big\}dS'\Big\}, r \in S. \end{aligned} \tag{11}$$

### 2.2. Near-Field Correction

Under the condition of far field, utilizing the Taylor expansion, the free space Green's function is approximated by

$$G\left(r,r'\right) \approx \frac{\exp\left(jk\hat{r}\cdot\left(r - r'\right)\right)}{4\pi r}. \tag{12}$$

where $r$ and $r'$ denote observation and source points, and $\hat{r}$ represents the unit vector of observation point.

However, the simplification of Green's function is unqualified in the near-field zone. Moreover, a minimum range beyond the Fraunhofer distance of $2D^2/\lambda$ must be satisfied to ensure the accurate representation of Green's function, where $D$ is the maximum dimension of the target [19]. However, as the size of the target increases, and considering the coupling scattering between multiple targets, it is challenging to meet the far-field region criterion. Therefore, the phase approximation of Green's function in the far-field assumption should be replaced to be more suitable for the near field. To modify Green's function for near-field scattering between multiple hybrid dielectric and conductor targets, an expansion center $r_n$ lying in the neighborhood of the source point is introduced [16,20], which is shown in Figure 1.

$$\left|r - r'\right| = \left|r - r_n - r' + r_n\right| = \left|(r - r_n) - \left(r' - r_n\right)\right| = |p - p'| \tag{13}$$

where $p = r - r_n$ and $p' = r' - r_n$. Due to the expansion center lying in the neighborhood of the source point, when the facet meshes densely enough, $|p'| \to 0$ and the proposed simplification can be qualified.

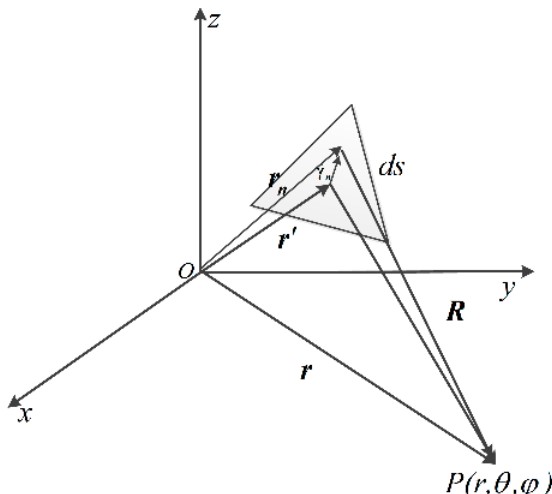

**Figure 1.** Schematic diagram of the facet and expansion center.

Equation (13) can be expressed as

$$\left| r - r' \right| = |r - r_n| \sqrt{ 1 + \frac{\left| r' - r_n \right|^2 - 2(r - r_n) \cdot (r' - r_n)}{|r - r_n|^2} } \tag{14}$$

when

$$\frac{\left| r' - r_n \right|^2 - 2(r - r_n) \cdot (r' - r_n)}{|r - r_n|^2} \to 0 \tag{15}$$

$\left| r - r' \right|$ can be approximated by $r - \hat{r} \cdot r'$.

Therefore, the near-field Green's function can be written as

$$G\left( r, r' \right) \approx \frac{\exp\left( jk\hat{\gamma}_n \cdot (r - r') \right)}{4\pi |\gamma_n|}. \tag{16}$$

where the unit vector $\hat{\gamma}_n$ can be defined as

$$\hat{\gamma}_n = \frac{r - r_n}{|r - r_n|} \tag{17}$$

Based on the aforementioned correction method, $J_i^{(k+1)}$ and $M_i^{(k+1)}$, which are the electric and magnetic current on facet $i$, respectively, can be precisely evaluated in (10) and (11).

### 2.3. Convergence Criterion

To terminate the update procedure within a limited period of time, the relative error during the $m$th iteration can be defined as

$$\alpha_m = \frac{\left\| \Delta J_i^{(m)} \right\|_2}{\left\| J_i^{(m)} \right\|_2} \times 100\%, \beta_m = \frac{\left\| \Delta M_i^{(m)} \right\|_2}{\left\| M_i^{(m)} \right\|_2} \times 100\%. \tag{18}$$

The procedure terminates when $\alpha_m \leq 3\%$ *and* $\beta_m \leq 3\%$ or $\alpha_m \geq \alpha_{m+1}, \beta_m \geq \beta_{m+1}$, where $m$ demotes the number of iterations. If the iteration procedure fully converges, the

relative error $\alpha_m, \beta_m$ tend to zero in finite time [4]. Therefore, the $\boldsymbol{J}_i^{(m)}$ and $\boldsymbol{M}_i^{(m)}$ on facet $i$ can be obtained. It was found that a relative error threshold of 0.03 yields sufficient accuracy [21].

*2.4. Acceleration Strategy*

For a given observation point, in the evaluation of surface integrals in (10) and (11), the $\boldsymbol{J}_i^{(n)}$ and $\boldsymbol{M}_i^{(n)}$ on facet $i$ are approximately calculated over all of the other facets. In the proposed algorithm, the computationally expensive operations compute the matrix-vector, which are $O(N^2)$ operations. To reduce computational complexity, the multilevel fast multipole method and the fast far-field approximation (FaFFA) are introduced to accelerate the iteration process; these reduce the operational cost by an order of magnitude to $O\left(N^{3/2}\right)$ for the electrically large targets. Instead of computing the interaction between groups of source and observation points directly, it is evaluated indirectly through aggregation, translation, and disaggregation, as shown in Figure 2. Meanwhile, the acceleration strategy of parallel programming is introduced to reduce the overall CPU time.

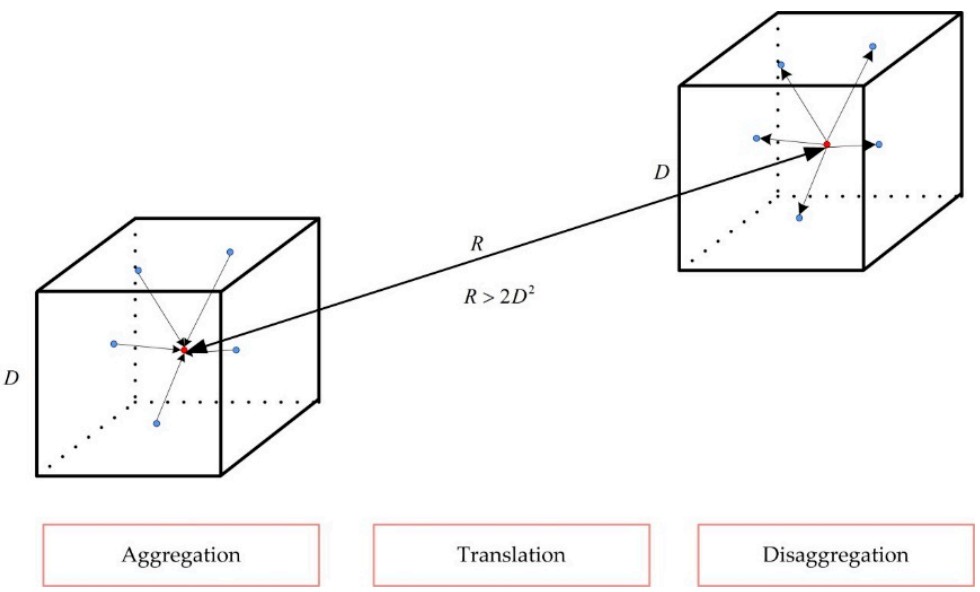

**Figure 2.** Far field with FaFFA acceleration, including aggregation translation and disaggregation.

**3. Numerical Results**

Several examples are presented in detail to show the efficiency and precision of the proposed method in this section. All of the examples here were run on an Intel(R) processor of 2.6 Ghz with 64 kernels and 128 GB RAM.

*3.1. Verification of the Method*

To validate the modified near-field IPO method, the EM scattering characteristic of multiple hybrid dielectric and conductor cubes with relative permittivity $\varepsilon_r = 4.4 - j0.22$ is analyzed in the following example. The working frequency is 1 GHz, and the plane wave illuminates the cubes in the far field along with the direction of $\theta_i : -90° \sim 90°, \varphi_i = 0°$.

In addition, the size of the cube is $D \times D \times D : 2\,\text{m} \times 2\,\text{m} \times 2\,\text{m}$, the sampling density is 8 facets per wavelength resulting in 30,096 triangles, and the distance between the two cubes is $R = 10\,\text{m}$. The location of the cubes is shown in Figure 3. It is apparent that the Fraunhofer distance of $R > 2D^2/\lambda$ is not satisfied. In order to ensure accurate iteration, it is vital to utilize the modified Green's function for near-field scattering between multi-hybrid dielectric and conductor cubes. For comparison, the results are shown in Figure 4, along with a reference computed by the MLFMM solver based on the numerically exact method of moment, with the data measured and processed through a compact range measuring

system by our research group. It can be seen that good agreement is observed between the results in most observation angles. The near-field correction improves the accuracy of the IPO method by compensating for the interaction between patches in the near field.

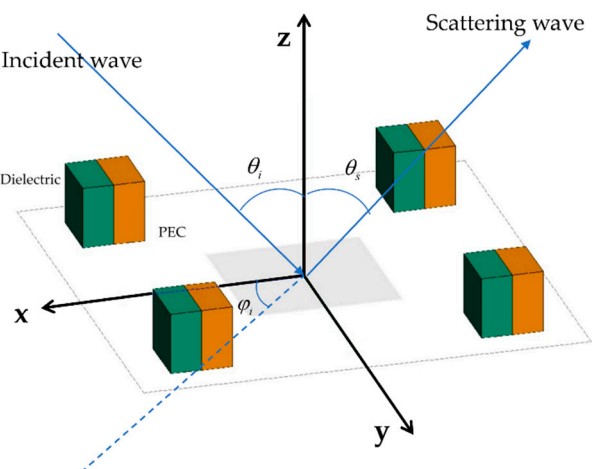

**Figure 3.** The location of four hybrid dielectric and conductor cubes in free space.

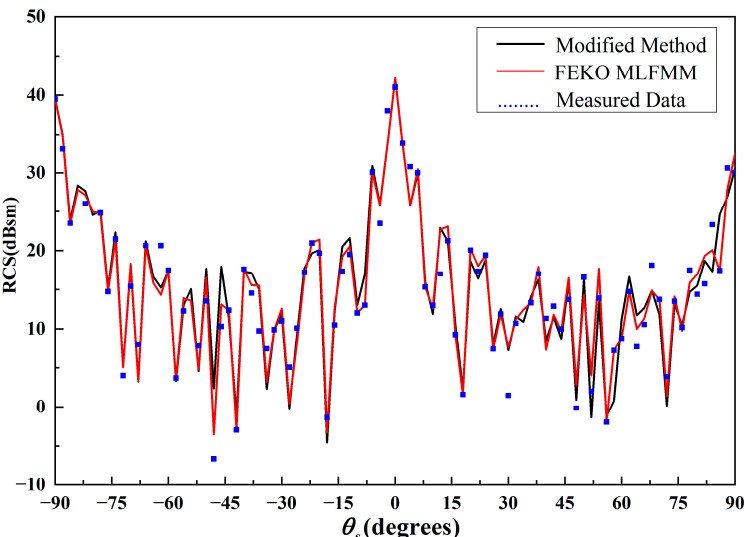

**Figure 4.** Mono-static RCS for HH polarization of four hybrid dielectric and conductor cubes in free space.

The MLFMM solver solution based on an accurate numerical method is more accurate compared with other high-frequency or hybrid methods for this geometry. Therefore, it can be utilized as a reference. It should be noted here that the major error results from edge scattering and creeping wave scattering, which are not considered in the proposed method. The memory requirement and CPU time for this example and root mean square error (RMSE) are given in Table 1.

The RMSE for whole observation angles can be expressed as

$$\text{RMSE} = \sqrt{\frac{1}{N}\sum_{n=1}^{N}\left(\sigma_n^{\text{MLFMM}} - \sigma_n^{\text{other}}\right)} \tag{19}$$

where $\sigma$ denotes the RCS at the observation angles.

**Table 1.** The RMSE and CPU time of the different methods.

| Method | RAM | CPU Time (s) | RMSE |
|---|---|---|---|
| Proposed Method | 122.414 | 1700 | 1.8423 |
| MLFMM Solver | 61.909 | 20,883 | 0.0000 |
| Measured Data | — | — | 2.2556 |

As we can see, the computing time of near-field IPO is about 8% of MLFMM. However, the memory requirement is two to three times higher than the MLFMM solver due to the parallel multicore programming strategy. Therefore, it can be concluded that the proposed solution increases the speed of calculation and saves CPU running time while increasing memory consumption as compared with the MOM-based MLFMM solver in commercial software FEKO.

### 3.2. Scattering Characteristic of Hybrid Dielectric and Conductor Cube Target

In the following example, the EM scattering characteristic of multiple hybrid dielectric and conductor cubes with relative permittivity $\varepsilon_r = 4.4 - j0.22$ is analyzed. The working frequency is 1 GHz, and the plane wave illuminates the cubes in the far field along with the direction of $\theta_i : -90° \sim 90°, \varphi_i = 0°$. In addition, the size of the cube is $D \times D \times D$ : 2 m $\times$ 2 m $\times$ 2 m, and the sampling density is 8 facets per wavelength. Figures 5 and 6 show the comparison results of dielectric cubes and hybrid dielectric and conductor cubes, respectively.

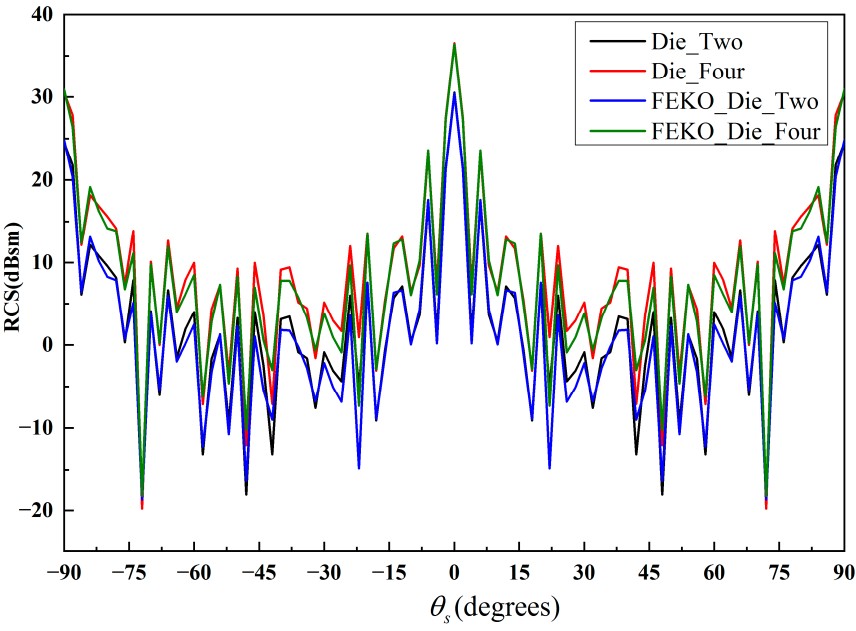

**Figure 5.** Mono-static RCS for HH polarization of dielectric cubes in free space.

As shown in Figures 5 and 6, good agreement is observed between the results calculated by the near-field IPO method and FEKO MLFMM. With the increase in the number of cubes, the mono-static RCS is 5 dB larger in most scattering directions. Meanwhile, the mutual iteration with different cubes in the near field becomes stronger. Moreover, due to the existence of refraction and transmission of the incident EM wave in the dielectric region, the energy of the scattered electric field from the dielectric cubes is weakened in all scattering directions.

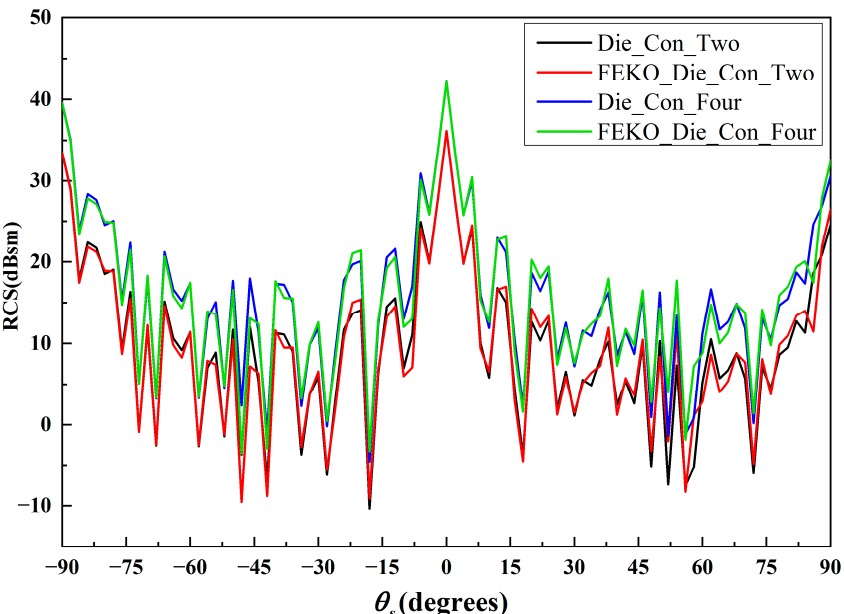

**Figure 6.** Mono-static RCS for HH polarization of different numbers of hybrid dielectric and conductor cubes in free space.

### 3.3. Scattering Characteristic of Missile-like Target

In the following example, the EM scattering characteristic of a missile-like target with dielectric parts is analyzed. The size of the target is $D_l \times D_w \times D_h$: 5.2 m × 2.77 m × 1.04 m. The geometry is discretized by a triangle mesh, resulting in 74,606 elements (74,606 flat, 0 curvilinear) corresponding to about 8 facets per wavelength. In addition, the distance between two targets is $R = 10$ m, and the Fraunhofer distance of $R > 2D^2/\lambda$ is not satisfied. The relative permittivity of the dielectric part, including the radome, wings, and empennage, is $\varepsilon_r = 4.4 - j0.22$. The geometry structure of the hybrid dielectric and conductor missile-like target is shown in Figure 7. The missile-like targets are parallel to each other side by side, and they head towards the positive direction of the *x*-axis.

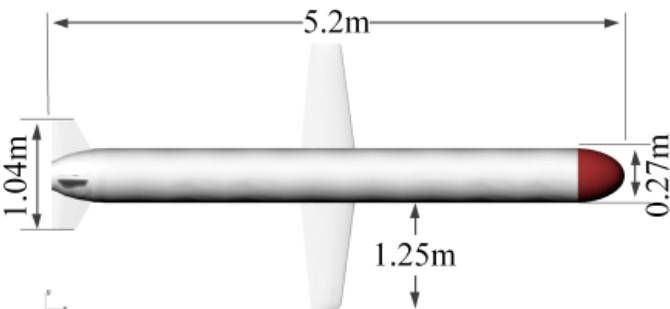

**Figure 7.** Geometry structure of the hybrid dielectric and conductor missile-like target.

Figure 8 presents the comparison of composite scattering of missile-like targets with different numbers. It shows that the mono-static RCS increases by about 6 dB in all scattering directions as the number of targets increases. Moreover, the trend of the curve does not change. However, as Figure 9 shows, with different distributions of two targets in space, the total composite scattering among all directions changes slightly. This shows that it is insensitive to the differences in distribution position. Therefore, the HH polarization RCS is insufficient to detect non-cooperative targets, especially when the target number is the same while the relative location is different.

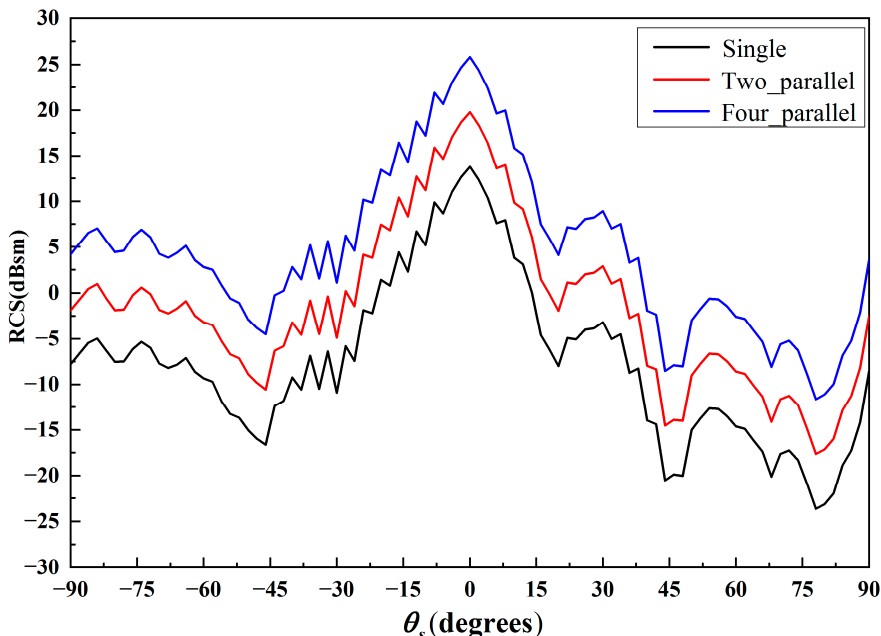

**Figure 8.** Mono-static RCS for HH polarization of hybrid dielectric and conductor missile-like targets with different numbers in free space.

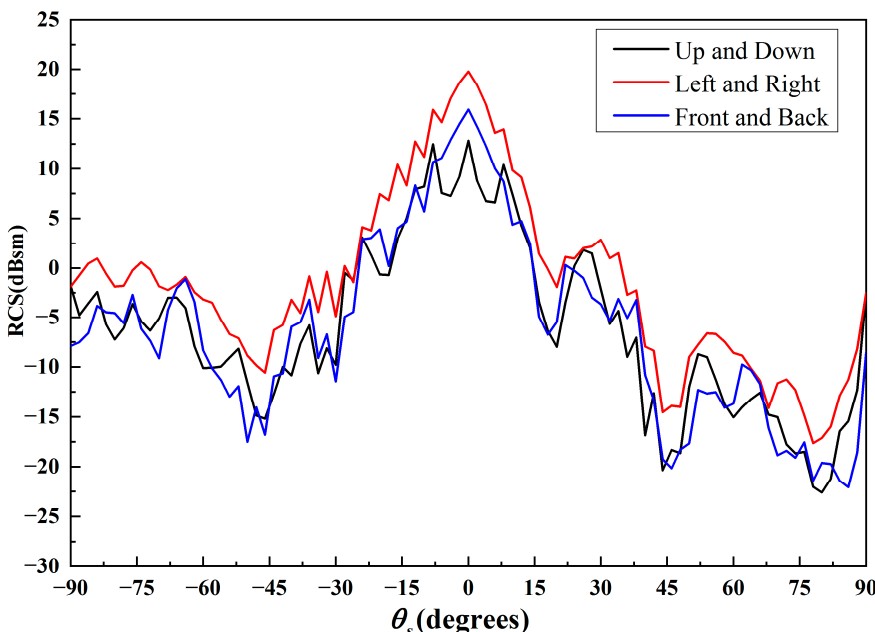

**Figure 9.** Mono-static RCS for HH polarization of hybrid dielectric and conductor missile-like targets with different distributions of position in free space.

*3.4. Scattering Characteristic of Missile-like Target above Sea Surface*

Figure 10 gives the schematic illustration of two missile-like targets above a rough sea surface. The rough surface is generated by the PM spectrum. The incident angle is $\theta_i = 60°$ and $\varphi_i = 0°$, while the observation direction ranges from $-90°$ to $90°$ at $\varphi_s = 0°$. The size of the rough sea surface is $L_x \times L_y$ : 15 m × 15 m, with four sampling points considered per wavelength. The wind speed at the height of 19.5 m above the sea surface is 1 m/s, 3 m/s, 5 m/s, and 9 m/s. The relative permittivity is $71.5 - j20$ when the incident frequency is 1 GHz, the water temperature is 23°C, the salinity of the sea water is 3.5‰, and $\alpha$ is 0.012 in the double Debye law. The targets are located at the height of 3 m above sea surface.

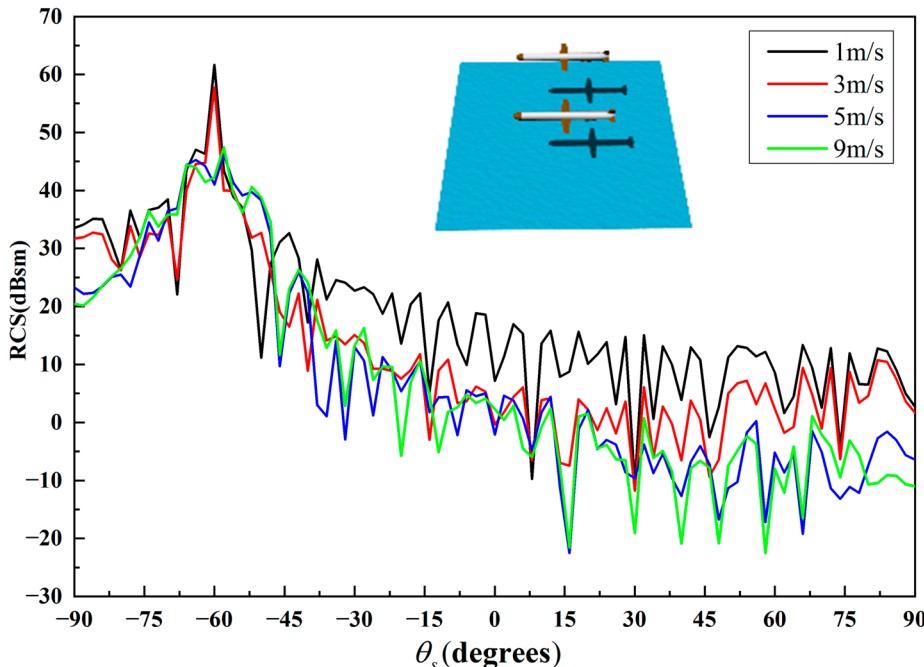

**Figure 10.** Bistatic RCS for HH polarization of hybrid dielectric and conductor missile-like targets with different wind speeds above the sea surface.

The scattering from the sea surface and the coupled scattering between the sea surface and the target are calculated by the proposed algorithm. Figure 10 compares bistatic RCS for HH polarization of hybrid dielectric and conductor missile-like targets with different wind speeds above the sea surface. It is observed that the bistatic RCS reaches a maximum at $\theta_s = -60°$ (the specular direction) and decreases with the increase of the wind speed at the height of 19.5 m above the sea surface. However, it increases in other scattering angles due to the diffuse reflection increasing with the increase of roughness. This is because the roughness of the sea surface will increase with the increase in wind speed. Meanwhile, when the wind speed reaches a particular value (5 m/s), the bistatic RCS changes slightly with the increase of the wind speed.

## 4. High Resolution Range Profiles

The distribution of the broadband radar echo signal in the radar line of sight represents the high resolution range profile with certain undulation patterns. HRRP is an efficient method for identifying different targets, since it is easy to obtain, easy to store, and contains rich structural features of the target. According to the scattering center model, it is known that the echo signal is a vector superposition of the sub-echo signals of each scattering center, and the phase difference is related to the relative position between the scattering points. Therefore, it will cause a change of the phase relationship between the sub-echo signals, which is expressed as the amplitude, translation, and attitude sensitivity of the range image.

### 4.1. HRRP Analysis of Missile-like Target

To improve the performance of non-cooperative target detection, the high-range resolution profiles of two missile-like targets with different distribution of positions in the free space are simulated. In this example, the frequency of the incident wave is 1 Ghz, and the incident angle is $\theta_i = 60°$ and $\varphi_i = 0°$. By analyzing the amplitude and location of the main scatter center, which corresponds to the main parts of the target, the location distribution of the two missile-like targets can be clearly distinguished.

It can be seen in Figure 11 that the length of the target in the radar line of sight is about 4.8 m. When targets are parallel and at the same height, there are four scatter centers, and the amplitude value increases obviously compared to the other two cases. However, while the targets are in a tandem array, there are two repeated sections of curves.

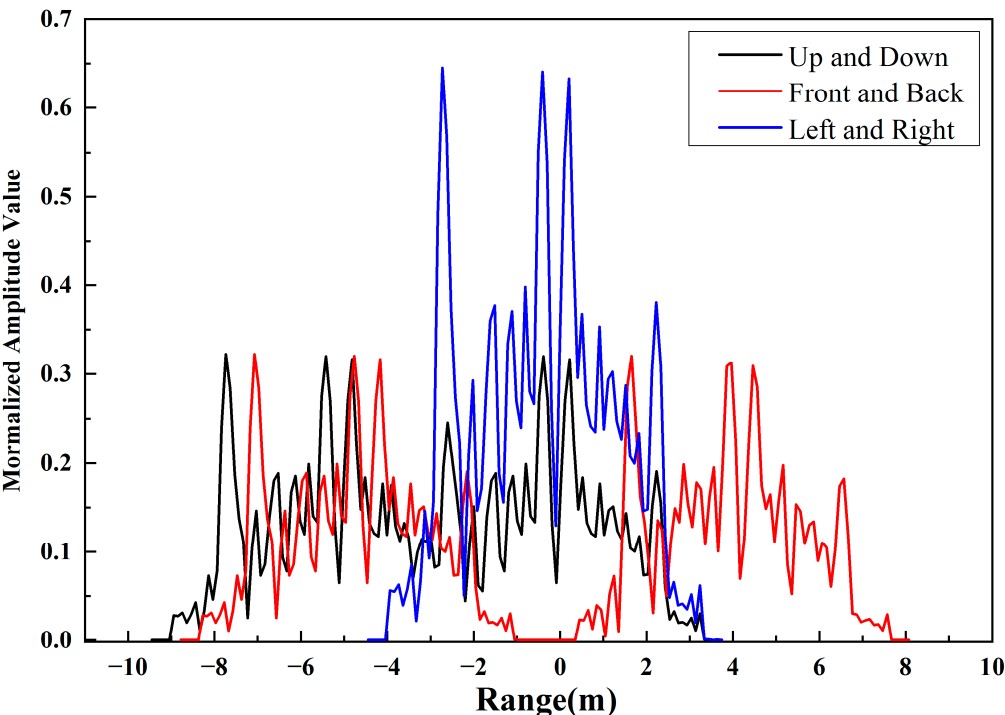

**Figure 11.** The HRRP of hybrid dielectric and conductor missile-like targets with different distribution of positions in free space.

Similarly, when the targets are parallel but at different heights, there are two repeated curves, but the amplitude value of the second section of the curve is smaller because of the partial occlusion in radar LOS. Meanwhile, the distance between the targets is shorter. Through analyzing the HRRP, the number of targets, the projection length, and the intensity of scattering can be determined, which can improve the performance of target detection and recognition.

*4.2. HRRP Analysis of Different Pitch Angles*

In this example, the targets are in a tandem array with the distance of 10 m. To analyze the effect of different pitch angles on the images, the pitch angle is set as $\theta_i = 65°$, $\theta_i = 75°$, and $\theta_i = 85°$, at $\varphi_i = 0°$. The frequency of the incident wave is 1 Ghz.

As shown in Figure 12, with the increase of the pitch angle, the normalized amplitude value of each strong scattering center decreases significantly. In addition, the scatter coefficient of the radome part decreases even less compared with that of the wings and empennage parts due to the different structural features of different parts. As is shown in Figure 7, the geometry of the wings and empennage part is flat; therefore, as the incident angle increases, the backward scattering decreases significantly. Because the radome part is an ellipsoidal-like structure, its scattering coefficient changes slightly with the variation of the pitch angle.

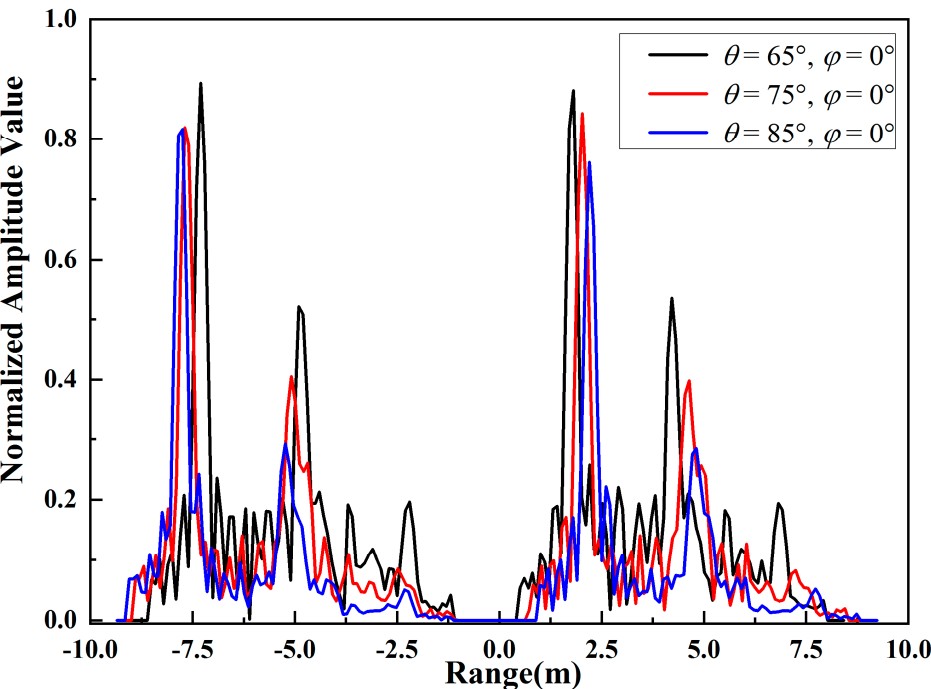

**Figure 12.** The HRRP of hybrid dielectric and conductor missile-like targets with different pitch angles in the free space.

The distance between the strong scattering center increases with the increase of the incident angle, because in the radar line of sight, the projection length of the distance between the strong scattering centers on the target becomes longer. This is manifested by the presence of extension in the *x*-axis direction and the overall translation toward the origin of the coordinate axis of the curve results. The original HRRP samples, as recognized in practical engineering applications, are translation sensitive, which will cause certain difficulties for subsequent recognizer design and model training; therefore, it is necessary to transform the position of the curves in the images.

### 4.3. HRRP Analysis of Different Azimuth Angles

In this example, the targets are in a tandem array with a distance of 10 m. To analyze the effect of different azimuth angles on the images, the pitch angle is set as $\varphi_i = 0°$, $\varphi_i = 15°$, and $\varphi_i = 30°$, with $\theta_i = 75°$. The frequency of the incident wave is 1 Ghz.

As shown in Figure 13, with the increase of the azimuth angle, the normalized amplitude value of each strong scattering center decreases significantly. In addition, the scattering coefficient of the radome part decreases even less compared with that of the wings and empennage parts due to the different structural features of different parts. In addition, due to the occlusion of the target, the amplitude value of the latter target changes less.

Furthermore, with the increase of the azimuth angle, the scattering coefficient of the second strong scattering center will decrease and then increase, and that of the third strong scattering center will increase and then decrease. It can be seen that the HRRP is sensitive to the changes of the incident angle, which will increase the difficulty of the feature extraction in the following target detection and recognition process.

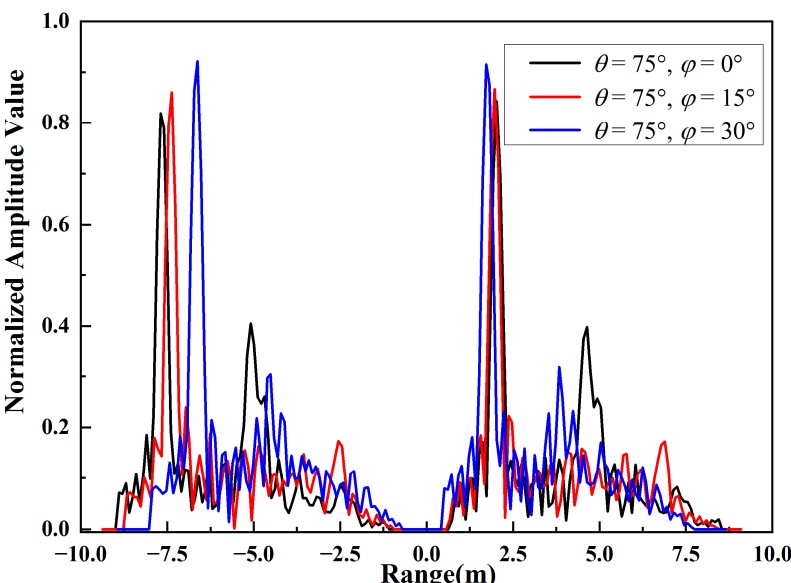

**Figure 13.** The HRRP of hybrid dielectric and conductor missile-like targets with different azimuth angles in the free space.

*4.4. HRRP Analysis of Different Incident Frequency*

In this example, the targets are in a tandem array with a distance of 10 m. To analyze the effect of different incident frequencies on the images, the incident frequency is set as 1 Ghz, 5 Ghz, and 10 Ghz with the incident angle at $\theta_i = 75°$, $\varphi_i = 0°$.

As shown in Figure 14, with the increase of the incident frequency, the normalized amplitude value of each strong scattering center changes differently. The scattering coefficient of the radome part and empennage part barely changes, while that of the wings and main body part of the target decreases rapidly with the increase of the incident frequency. As a result, the scattering center can be difficult to extract. Therefore, with the changes in radar operating frequency, the HRRP results of the same target change greatly, and this will increase the difficulty of the feature extraction in the following target detection and recognition process.

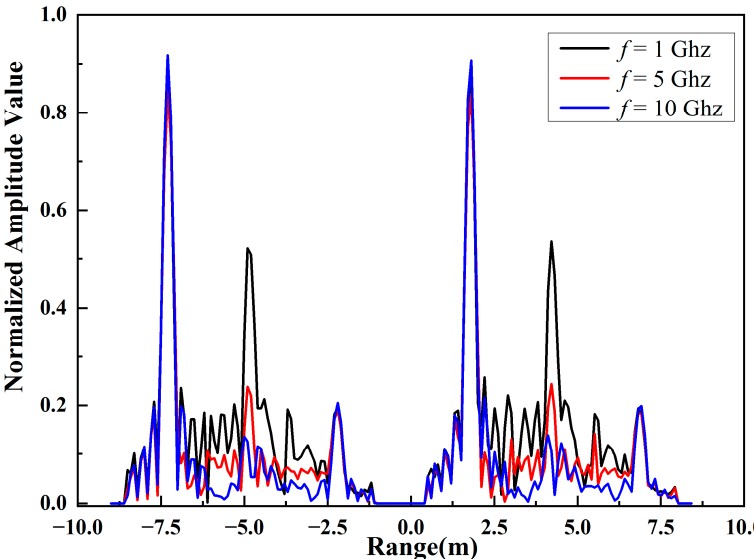

**Figure 14.** The HRRP of hybrid dielectric and conductor missile-like targets with different incident frequency in the free space.

### 4.5. HRRP Analysis of Different Target Types

To analyze the HRRP of different target types, the simulation of three different missile-like targets is conducted in the following example. The geometry structure of different missile-like targets is shown in Figure 15 [17]. The size of the type 1, type 2, and type 3 targets is 2.09 m × 0.38 m, 4.63 m × 0.62 m, 8.60 m × 1.88 m, respectively. The working frequency is set as 1 Ghz with the incident angle at $\theta_i = 75°$, $\varphi_i = 0°$. The targets are in a tandem array with a distance of 10 m. The comparison of HRRP from different types of targets is presented in Figure 16.

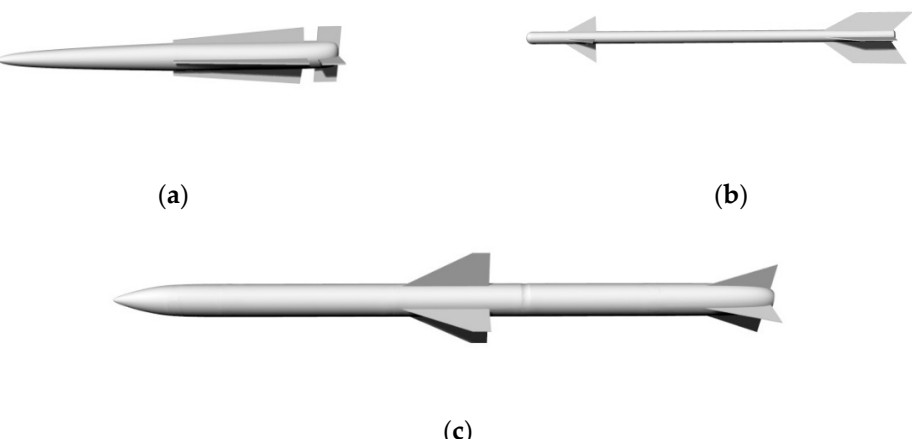

(**a**)　　　　　　　　　　　　　　　(**b**)

(**c**)

**Figure 15.** The geometry structure of different types of missile-like targets: (**a**) type 1; (**b**) type 2; (**c**) type 3.

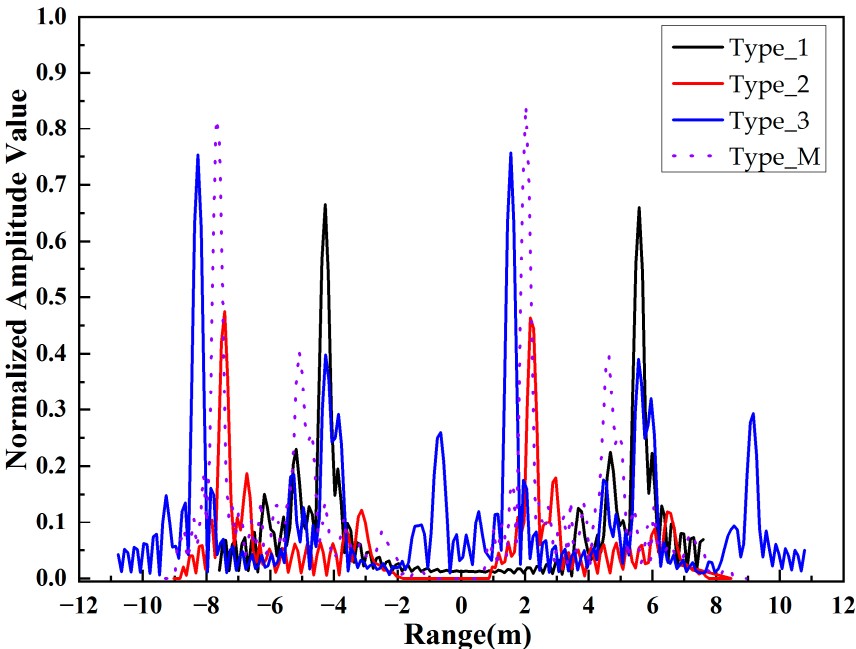

**Figure 16.** The HRRP of different types of hybrid dielectric and conductor missile-like targets in the free space.

As shown in Figure 16, with different geometry structures, the number, relative position, and amplitude value of the main scattering centers on different targets are different from each other. Therefore, an HRRP dataset of different targets under different conditions can be established through simulation experiments. Moreover, based on the deep learning convolution neural network model combining improved methods, an automatic target

detection and recognition model can be obtained, which can significantly promote the accuracy and efficiency of target detection and recognition.

## 5. Discussion

In this section, the results of the experiments presented in Sections 3 and 4 are discussed in detail. First, the electromagnetic scattering characteristics of hybrid dielectric and conductor missile-like targets are discussed. Then, based on the proposed electromagnetic scattering computation method, the HRRP of different targets under different conditions is obtained; the second part includes HRRP analysis and discussion.

### 5.1. Scattering Characteristics

To validate the proposed method, the numerical simulation results are compared with the method of moment-based MLFMM solver in commercial software FEKO and the measured data. In contrast to the original iterative physical optics method, the proposed method considers the mutual effects between each facet. Therefore, a good agreement can be seen between the results in most ranges of angles. Then, electromagnetic scattering characteristics of multiple hybrid targets are simulated and analyzed. It can be seen that with the increase of the target number, the RCS increases. However, the total composite scattering coefficient among all directions changes slightly when the number of targets is the same while the relative position distribution is different. This causes great difficulty in target detection and recognition.

### 5.2. HRRP Analysis

To fully exploit the scattering information, the HRRP is analyzed in Section 4. Compared with the RCS results mentioned above, the range profile of targets with different location distributions can be easily distinguished. Although the HRRP results are sensitive to the incident angles and incident frequency, they can be used to detect and recognize the target, especially with a large number of simulation results. Furthermore, the HRRP of different targets under different conditions is simulated for the construction of a feature dataset, which can be used for automatic target detection and recognition without high human consumption and computation time cost.

## 6. Conclusions

In this paper, the far-field EM scattering characteristic of multiple hybrid dielectric and conductor targets is analyzed considering the near-field iteration procedure. The near-field IPO is utilized to calculate the interaction between different targets and the rough surface. To accelerate the iteration process, MLFMM, FaFFA, and parallel programming are introduced. Finally, the mono-static and bi-static RCS of multiple hybrid dielectric and conductor targets under different conditions is obtained. The results will have critical applications in radar detection, remote sensing, etc. However, the HH polarization RCS is insufficient to detect non-cooperative targets, especially when the number of targets is the same while the relative location is different. Therefore, in the future, we will address composite scattering, HRRP, and SAR imaging of multiple hybrid dielectric and conductor targets.

**Author Contributions:** Conceptualization, C.T. and X.L.; methodology, Y.W., Q.W. and Z.W.; project administration, C.T.; resources, X.L.; software, Q.W.; validation, Y.W.; writing—original draft, Q.W.; writing—review & editing, Y.W., Z.W. and T.W. All authors have read and agreed to the published version of the manuscript.

**Funding:** This research was funded by the National Natural Science Foundation of China, grant number 62201611, and the Natural Science Foundation of Shannxi Province, grant numbers 2021JQ-362 and 2021JQ-365.

**Data Availability Statement:** Not applicable.

**Acknowledgments:** The authors are grateful to Chuangming Tong for his technical assistance throughout the project and would like to acknowledge the contributors who provided important feedback on the research.

**Conflicts of Interest:** The authors declare no conflict of interest.

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
