# Peer review of "Near-Field IPO for Analysis of EM Scattering from Multiple Hybrid Dielectric and Conductor Target and High Resolution Range Profiles"

_remotesensing, doi:10.3390/rs15071884_

Round 1
Reviewer 1 Report (Previous Reviewer 2)
The authors have adequately responded to my criticisms, therefore I suggest accepting the article for publication in Remote Sensing.
Author Response
Please see the attachment.

Reviewer 2 Report (Previous Reviewer 3)
The paper deals with the computation of the scattering from multiple targets which are in the far field with respect to the antennas but in the near field one to the other. The authors claim in the abstract that the paper deals with targets over a rough surface. But they present only one numerical example related to multiple targets over a rough surface, without explaining how the surface is considered in the algorithm. The abstract should be more precise about goals and results.
Green’s function is provided in eq. (3). It is repeated with a different expression in Eq.(12), where some symbols are not defined (r and hat r).
The rationale for the numerical examples is not given. The reference system is not provided, so the figures' meaning is unclear (the x,y,z axes are not indicated, and neither the two angles theta and phi).
Moreover, it is not clear if the multiple interactions between the targets are actually significant. Looking at the results in Fig. 8, it seems that the multiple targets case is just the simple summation of the single ones.
In example 3.4, how the sea surface is treated in the algorithm is not explained. The targets in this last example are moving, but in the problem formulation there is not a trace of the target’s speed. So, how it is inserted into the algorithm?
References are at most conferences. It would be preferable to replace some of them with journal papers.
Author Response
Please see the attachment.

This manuscript is a resubmission of an earlier submission. The following is a list of the peer review reports and author responses from that submission.
Round 1
Reviewer 1 Report
This paper presents a near-field iterative physical optics (IPO) method based on a modified near-field Green’s function for the far-field composite electromagnetic (EM) scattering, which is suitable for rough surface and multiple hybrid dielectric, conductor targets above it. In this paper, the near field correction is applied to improve the accuracy and the parallel computation is used to improve the efficiency. Besides, the characteristics of multiple hybrid dielectric and conductor targets target range image are discussed in detail. The presented methodology is novel and has significance. However, here are some questions better be answered or explained before it is considered for publication.
1. Why does the simulation RCS result curve appear similar, as the number of targets increases in Fig.8? In addition, is the interference between echoes of different targets considered?
2. How to determine the peaks corresponding to wings and empennage parts respectively, when analyzing the spatial distribution of missile-like target’s HRRP. And how to establish the connection between the simulation peak location and target’s local structure?
3. What is the physical basis for setting the value of the relative permittivity of sea surface?
4. Some details on computational times better be introduced in this context. What is the overhead to compute the EM scattering characteristic of complex targets?
Reviewer 2 Report
Review of “Near-field IPO for Analysis of EM Scattering from Multiple Hybrid Dielectric and Conductor Target and High Resolution Range Profiles” by Qingkuan Wang et al.
The authors provide a method based on near-field iterative physical optics and a modified near-field Green’s function for the far-field composite electromagnetic scattering analysis. In particular, the Authors, apply this method to analyse the electromagnetic scattering of a rough surface and multiple hybrid dielectric and conductor targets above it in the near-field region.
The authors provide both mathematical and numerical evidence in support of the thesis, showing several results on different scenario (different targets under different conditions).
I encountered very strong critical issues that force me to refuse the publication of the paper in Remote Sensing.
Main criticism
The first somewhat regrettable thing is the following statement by the authors:
“No IPO method specifically suitable for the near-field EM scattering of multiple hybrid dielectric and conductor targets has been developed. Based on the modified near-field Green’s function, we presented a near-field IPO method for the far-field composite EM scattering analysis of rough surface and multiple hybrid dielectric and conductor targets above it in the near-field region.”
But an entirely analogous method was published by the same authors about 6 months ago [Wang et al. "Composite Electromagnetic Scattering and High-Resolution SAR Imaging of Multiple Targets above Rough Surface," Remote sensing 2022, 14, 2910.]
This fact struck me (negatively) twice, one because of the above statement and second because this article is never referred to in this manuscript.
The work presented seems to me to be fairly incremental work. The two manuscripts share a large part of the theoretical and numerical study.
The study of numerical convergence and the acceleration strategy have been added. Obviously, the studied targets have different shapes.
For these reasons I consider this paper an incremental work that cannot be accepted by Remote Sensing.
Reviewer 3 Report
The paper content is presented in a rather confused style. English is poor; therefore, it hinders comprehension. The goal of the paper is not clearly stated in the abstract.
The text is not adequately formatted. Moreover, different quantities are indicated by the same symbols. For instance, the relative errors and the permittivity and conductivity; also, the subscript “i” is used to indicate both incident fields and facets. In Eq (13) the plain symbol p stands for a vectorial value (vectors are indicated in bold in the other equations). The Green function in (16) is not coherent with (12) and definition (17). There are many symbols not defined.
Beyond the merely formal aspect, the whole section 2 is a generic presentation of well-known equations, and it does not provide significant insight into the proposed method and how it works.
In conclusion, the paper in its current form does not provide sufficient elements to be evaluated.
I reserve to provide a more detailed review of the technical aspects in a possible further evaluation phase.
